# The Fusion of ERA5 and MERRA-2 Atmospheric Temperature Profiles with Enhanced Spatial Resolution and Accuracy

Yale Qiao [1,2], Dabin Ji [1], Huazhe Shang [1,*], Jian Xu [3], Ri Xu [1,4] and Chong Shi [1]

1   State Key Laboratory of Remote Sensing Science, The Aerospace Information Research Institute, Chinese Academy of Sciences, Beijing 100101, China; 1020201644@glut.edu.cn (Y.Q.); jidb@aircas.ac.cn (D.J.); 2020120155@nwnu.edu.cn (R.X.); shichong@aircas.ac.cn (C.S.)
2   College of Geomatics and Geoinformation, Guilin University of Technology, Guilin 541004, China
3   National Space Science Center, Chinese Academy of Sciences, Beijing 100190, China; xujian@nssc.ac.cn
4   College of Geography and Environmental Science, Northwest Normal University, Lanzhou 730070, China
*   Correspondence: shanghz@radi.ac.cn

**Abstract:** Accurate high-resolution atmospheric temperature profiles are essential for precisely characterizing the evolution of the atmosphere and developing numerical forecasts. Atmospheric datasets, such as ERA5 (the fifth-generation ECMWF Reanalysis) and MERRA-2 (the Modern-Era Retrospective Analysis for Research and Applications, Version 2), provide global and continuous temperature profiles, with fine vertical distribution and horizontal resolution. RAOB (Radiosonde Observation) sounding data have high confidence and representativeness and are usually used for data accuracy verification. Due to the difficulty of updating existing products, and the scarcity of research on mesospheric temperature profiles, this work maximizes the high observation accuracy of RAOB data, combines the benefits of ERA5's horizontal resolution and MERRA-2's vertical distribution, and employs the optimal interpolation method to combine the data, in order to produce a fused result with high spatial resolution. After converting all of the data to the same spatial distribution, the optimal interpolation method was used to combine the two datasets from separate places and different pressure layers in order to produce the fused results, which had a vertical distribution of 45 layers and a spatial resolution of 0.25°. The fused data's RMSE and MAE were 6.0 K and 5.0 K lower than those of the MERRA-2 temperature profile data, respectively, and 0.3 K and 0.4 K lower than those of the ERA5 temperature profile data, respectively. The validation, using data from 2019, showed that the fused data exhibits better correlation and data accuracy than the other two datasets, which demonstrated that the fused algorithm can potentially be used to generate reliable datasets for future meteorological research.

**Keywords:** temperature profiles; fused data; optimal interpolation; global scale; ERA5; MERRA-2; RAOB

## 1. Introduction

Temperature profiles represent the temperature values at different altitudes, which is an essential meteorological characteristic of the atmosphere and the basis for accurately retrieving atmospheric trace gases. Meanwhile, temperature profiles represent the thermal conditions of the atmosphere, and monitoring their distribution and variability is crucial for researching climatic phenomena, such as the greenhouse effect [1]. Since studying climate radiative forcing requires global-scale observations of temperature, the accurate research and descriptions of atmospheric evolution, climate change, numerical forecasting, short-range warning, artificial weather impact, and other meteorological protection efforts rely on the real-time and effective detection of atmospheric temperature profiles.

In 2008, China established the National Key Science and Technology Infrastructure Project, "Integrated Ground-based Monitoring Meridian Chain for the Eastern Hemisphere Space Environment", in order to establish comprehensive observatory stations for continuous monitoring of the middle and upper atmosphere from the Earth's surface [2]. The

middle atmosphere is a crucial aspect in the transition to the space environment between the Sun and the Earth. The phenomenon of temperature inversion in the middle atmospheric layer has a significant impact on the launch probability of missiles and satellites, the probability of accurate orbital access, and the operational lifetime of such technology; therefore, monitoring the temperature of the middle layer is of great importance for scientific research and aerospace activities [3]. The mesosphere evolves more slowly than the troposphere, and its downward transport may lead to continuous and predictable changes at the surface it. Thus, is also important to observe or predict the temperature of the mesosphere for weather studies involving the troposphere or surface [4].

Currently, there are four primary sources of atmospheric temperature profiles: satellite-based measurements, airborne measurements, ground-based measurements, and reanalysis data. Among them, satellite-based measurements provide a wide range of observations and are continuous on the time scale, but their observation time spans are shorter compared to those of reanalysis data.

Between 2008 and 2021, China successfully launched five 'Fengyun-3' series meteorological satellites (FY-3A, FY-3B, FY-3C, FY-3D, FY-3E), with a microwave hygrometer, microwave thermometer, microwave imager, and other payloads on board [5,6]. However, FY-3A and FY-3B are currently out of operation. The Fengyun (FY)-3C satellite carries the microwave humidity and temperature sounder on board, which began taking measurements on 30 September 2013. The microwave humidity and temperature sounder (MWHTS) observes the vertical distribution of atmospheric temperature and moisture [7]. The Advanced Technology Microwave Sounder (ATMS), which is on board the Suomi NPP (National Polar-orbiting Partnership), the preparatory star for the next-generation U.S. polar-orbiting weather satellite JPSS (Joint Polar Satellite System), launched in 2011, is a combination of microwave thermometer AMSU-A and microwave hygrometer AMSU-B/MHS that detects the vertical distribution characteristics of atmospheric temperature and humidity [8].

Airborne measurement is expensive and lacks time continuity, so this measurement method is generally not used; for ground-based measurement, which includes measurements using a microwave radiometer, a sounding balloon, or other methods, a ground-based microwave radiometer measurement shows good time continuity for temperature profile observation, but it is greatly affected by weather: especially under cloudy conditions, the uncertainty of the cloud absorption coefficient leads to an increase in error or even failure. Conventional sounding measures are the most reliable and representative approach for measuring atmospheric temperature profiles, but they have limits in terms of temporal continuity, station spread, and expense. RAOB (Radiosonde Observation) sounding data are the actual measurements from radiosondes at global weather stations from the National Oceanic and Atmospheric Administration (NOAA)-National Environmental Satellite Data and Information Service (NESDIS) operational meteorological database archive [9], whose soundings have high confidence and representativeness and are generally used for data accuracy validation; for example, Ma, Y. et al. (2020) used RAOB data to verify the accuracy of atmospheric temperature data retrieved from AIRS for the Taklamakan Desert [10]. In addition, at present, some platforms in various countries use assimilation and other technologies to fuse satellite, ground-based, aircraft, ship, and other data in order to produce reanalysis data with good temporal continuity and wide spatial coverage. Reanalysis is the process of reprocessing a series of observational data using an assimilation system, which typically produces data with a wide variety of parameters, a long duration, and a wide spatial resolution, and the role of reanalysis in climate monitoring applications is now widely recognized. As a result, reanalysis data are increasingly used in the fields of agriculture, weather monitoring, energy, oceanics, etc. The temporal and spatial resolutions of various reanalysis datasets have been increasing, and the time span has increased from a decade to more than 100 years.

Data assimilation is a method of incorporating new observations into the dynamic operation of a numerical model, taking into account the spatial and temporal distributions

of the data and the errors in the observed and background fields, which is therefore equally applicable to data fusion. Data assimilation methods commonly include successive correction, optimal interpolation, 3D/4D variation, and Kalman filters. In order to understand 3DVar and 4DVar, it is necessary to understand Bayesian theory and great likelihood estimation, as well as some basic variational theories. If one wants to understand the ensemble Kalman filter, one needs to understand the ideas of theories such as minimum variance estimation [11,12]. The successive correction method and the optimal interpolation method are similar, in that they both make a distinction between the observed values and the points to be assimilated, which are then interpolated into the values of the points to be assimilated, and then results are finally obtained as analytical values. The difference between them is that, using the optimal interpolation method, the weight function is calculated by minimizing the analytical variance. Therefore, the biggest improvement in the optimal interpolation method compared with the successive correction method is that, when calculating the weights, the correlation between various observational errors and the correlation between different observations are considered. This avoids the disadvantage of arbitrary weight selection, as occurs when using the successive correction method [13,14]. The variational assimilation method uses the numerical model as a kinetic constraint. It reduces the data fusion to the problem of solving the extrema of the objective function, characterizing the deviations between the analyzed and observed fields and the background field [15]. If the objective function is defined in three dimensions (excluding the time dimension), then it is a three-dimensional variational method; if it is defined in four dimensions, it corresponds to a four-dimensional variational method. Due to the great computational effort required for the four-dimensional variational method, it is relatively rare in the operational application of data fusion. Based on the sounding data, Cai Yi et al. (2017) used the optimal interpolation method to correct the atmospheric profile of MODIS inversion. The profile accuracy can be effectively improved in areas with corresponding ground stations. However, the method is no longer applicable in places where ground stations are missing [16]. S. Mahagammulla Gamage et al. (2020) combined the measurements of Raman lidar with ERA5 data. For this approach, the authors used the one-dimensional variational method, based on the optimal interpolation method, to finally obtain the fused product, in which the initial separate products were improved to some extent [17].

Usually, the reanalysis data cover the whole world, such as with ERA5, MERRA-2, JRA55, NCEP, and other datasets, and they are better than sounding data in terms of spatial and temporal resolution, which makes them very suitable for analyzing spatial and temporal changes over long periods. Currently, data from the National Oceanic and Atmospheric Administration (NOAA)/National Centers for Environmental Prediction (NCEP) and the European Centre for Medium-Range Weather Forecasts (ECMWF) are available. The National Aeronautics and Space Administration (NASA) Global Modeling and Assimilation Office (GMAO) and the latest reanalysis programs of the Japan Meteorological Agency (JMA) additionally provide a rich collection of climate data products [18]. Among them, the temperature profiles of ERA5 and MERRA-2 have relatively high accuracy and are widely used. Robert M. Graham et al. (2019) used AWI radiosonde observations to assess the accuracy of five global atmospheric reanalysis datasets from the Fram Strait, including temperature profiles from ERA5, ERA-I, JRA-55, MERRA-2, and CFSv2. The ERA5 temperature profile demonstrates the best accuracy among all five reanalyses, with a correlation coefficient of 0.96 and a deviation of 0.3 °C from the actual value. In contrast, MERRA-2 exhibits the second-best accuracy, with a correlation coefficient of 0.95 and a deviation of 0.5 °C from the actual value [19].

Among the currently known temperature profile products, ERA5 has high temporal and horizontal resolutions, which can reach 1 h and 0.25°, respectively [20]. In terms of vertical distribution, MERRA-2 has the finest relative pressure distribution, at 42 levels [18]. Compared to other reanalysis data, JRA-55 has a temporal resolution of 6 h and a spatial resolution of 125 km. NCEP-DOE AMIP-II has a temporal resolution of 6 h and a spatial resolution of 250 km, both of which are relatively coarse in resolution; hence, ERA5 and MERRA-2 are used

for data fusion in this research [21,22]. Moreover, for some pressure layers unique to MERRA-2, most occur in the stratosphere and mesosphere. Although seasonal weather predictions were previously based mainly on data from the troposphere, meteorological data from the middle atmosphere are becoming increasingly non-negligible in climate change predictions. Since the turn of the 21st century, data modeling studies have increasingly incorporated stratospheric and even mesospheric information. While the spatial and temporal resolutions of ERA5 are high, it lacks some information about the mesosphere.

Therefore, in this paper, the optimal interpolation method with high efficiency is chosen for data fusion, mainly by taking full advantage of the high observational accuracy of RAOB sounding data, combining the advantages of the horizontal resolution of ERA5 and the vertical distribution of MERRA-2, and using the optimal interpolation method to optimally fuse the two data that ERA5 and MERRA-2 in order to avoid the problem of the discontinuity of sounding data. A spatially continuous fused product, with a horizontal resolution of 0.25°, a pressure layer of 45 layers, and high accuracy, is obtained. The final generated temperature profile fused data can be used for meteorological studies.

## 2. Data and Methods

### 2.1. Introduction of RAOB

Radio soundings (radiosonde) are devices placed on sounding balloons that can observe parameters such as wind direction, wind speed, temperature, humidity, and pressure at different altitudes, generally up to 30 km. They are the primary tools for high-altitude meteorological observations.

RAOB releases sounding balloons at fixed stations twice daily, at UTC 0000 and 1200. Any hourly data between UTC 0000 and 1200 are stored at the UTC 0000 location. Similarly, hourly data from UTC 1200 to 2300 are stored at the UTC 1200 location. Thus, for example, data from UTC 1500 will be stored at UTC 1200, and data from UTC 0000 and 1200 can be accessed and downloaded from https://ruc.noaa.gov/raobs/ (accessed on 20 June 2023) [23].

The RAOB observations mainly include parameters such as temperature, dew point drop (difference between temperature and dew point temperature), pressure, wind speed, and wind direction in different pressure layers, as well as relative humidity, which is mainly calculated using the temperature and dew point drop [24,25]. Pressure is mainly distributed among 22 pressure layers, but through the survey, most of the data in pressure layers above 10 hPa are invalid; therefore, only 16 of them are used in this paper: 1000, 925, 850, 700, 500, 400, 300, 250, 200, 150, 100, 70, 50, 30, 20, and 10 (hPa) [23,26]. Sounding data are essential and reliable measurements of atmospheric temperature and humidity profiles. The information used in digital weather forecasting nowadays is mainly obtained from the sounding station network [27], which has high representativeness and credibility and is generally used to obtain actual values with which to evaluate the accuracy of the temperature and humidity profile data inverse using bright temperature data. However, it has limitations in terms of time continuity, station distribution, and cost.

### 2.2. Introduction of Reanalysis Data

ERA5 is the fifth generation of global climate reanalysis datasets, released by ECMWF in 2017, and it includes reanalysis datasets from 1950 to the present, which uses a new generation of four-dimensional variational (4D-Var) assimilation technology, with components from more than 200 satellite instrument observations or conventional data types [20]. It currently provides hourly real-time atmospheric reanalysis data, including both pressure- and single-layer data, covering mainly wind, radiation, clouds, heat, soil, temperature, vegetation, snow, and other related parameters, with a horizontal spatial resolution of $0.25° \times 0.25°$, thus providing a complete set of meteorological data comparative to ERA-Interim (6 h, $1.5° \times 1.5°$). Users can access and download ERA5 data at https://cds.climate.copernicus.eu/cdsapp#!/search?type=dataset (accessed on 20 June 2023) [28]. For the ERA5 pressure reanalysis data, the pressures are mainly distributed at 1000, 975, 950, 925, 900, 875, 850, 825, 800, 775, 750, 700, 650, 600, 550, 500, 450, 400, 350, 300, 250,

225, 200, 175, 150, 125 100, 70, 50, 30, 20, 10, 7, 5, 3, 2, and 1 (hPa), in 37 pressure levels, which demonstrates a more refined pressure layer distribution compared to the sounding data [29]. ERA5 reanalysis data have been well used in studies of profile, precipitation, weather variability analysis, and wind energy resources [30].

MERRA-2 (Modern-Era Retrospective Analysis for Research and Applications, version 2) is a reanalysis product developed by NASA's Global Modeling and Assimilation Office (GMAO), spanning from 1980 to the present, using a three-dimensional variational (3D-Var) scheme within the Grid Statistical Interpolation (GSI) scheme, with a 3 h update cycle. Compared to MERRA, MERRA-2 includes additional satellite observations [31,32]. The resolution of this reanalysis data is $0.5° \times 0.625°$, and the pressure is mainly distributed at 1000, 975, 950, 925, 900, 875, 850, 825, 800, 775, 750, 725, 700, 650, 600, 550, 500, 450, 400, 350, 300, 250, 200, 150, 100, 70, 50, 40, 30, 20, 10, 7, 5, 4, 3, 2, 1, 0.7, 0.5, 0.4, 0.3, and 0.1 (hPa), in 42 pressure levels. Users can access and download MERRA-2 data at https://disc.gsfc.nasa.gov/ (accessed on 20 June 2023) [18].

### 2.3. Research Methods

#### 2.3.1. Data Fusion Methods

Gandin (1963) proposed the optimal interpolation method, based on statistical estimation theory, by introducing statistical methods [33]. The key to the optimal interpolation method is finding the weight matrix. If the background error covariance matrix can be reasonably estimated, then the optimal interpolation method can be implemented very simply. Therefore, in this paper, we chose the optimal interpolation method with relatively good accuracy and high efficiency for data fusion. The optimal interpolation method assumes that the background, observed, and analyzed values are unbiased estimates so that the analytical variance is minimized and, thus, the weight coefficients are obtained.

The principle of this method is to find the analytical value of each grid point in the space by adding the revised value to its initial value, which is obtained by weighting the deviation of the observed values around the grid point from the initial value. The following Equation (1) can express the analytical value of each grid point.

The specific steps in the calculation formula are as follows:

$$T_i^a = T_i^b + \sum_1^k p_i \left( T_i^{obs} - T_i^b \right) \equiv T_i^b + p_i^T \left( T^{obs} - T^b \right) \tag{1}$$

The principle of the optimal interpolation method is shown in Figure 1, using K observed temperature profiles in the range R to correct the initial temperature profile located at the center. This expression is shown in Equation (1). $T_i^b$ represents the temperature of the ith point to be fused, R represents a certain range, and $T_i^{obs}$ represents the ith observation in the range of R. In this paper, we selected eight pixel points around the point to be fused as the data in the range R for fusion. The meaning of fusion is to let the points be fused using the difference of the k observations within the range of R for weighted summation in order to obtain the final algorithm result, using this method to fuse the data from MERRA-2 and ERA5, where the expressions of the weights are obtained using the minimum variance estimation. Based on the data accuracy verification results in Section 3.1, the observations and the points to be fused are judged to be those of MERRA-2 or ERA5.

$$p_i = (a + b)^{-1} c_i \tag{2}$$

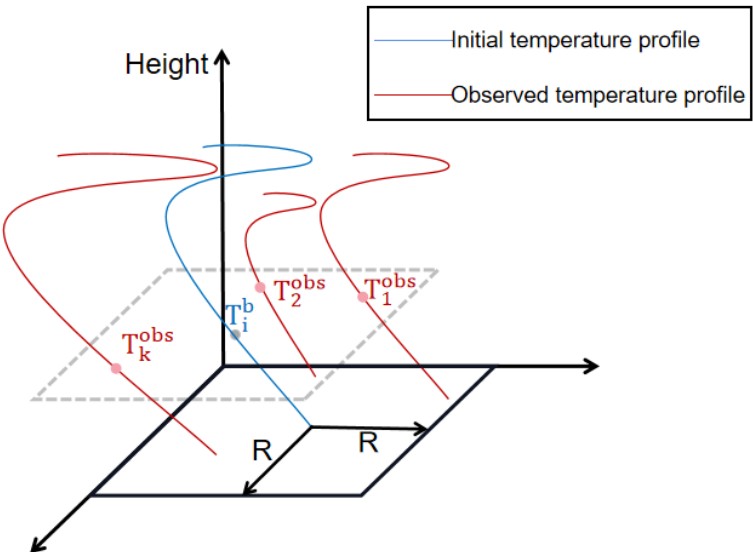

**Figure 1.** Principles of the optimal interpolation method.(R: Range of correction of the initial temperature profile at the center).

The weighting equation is calculated using the true, observed, and initial values, a is the background field covariance matrix.

$$
a = \begin{pmatrix}
\overline{\left(T_1^b - T_1^t\right)\left(T_1^b - T_1^t\right)} & \cdots & \overline{\left(T_1^b - T_1^t\right)\left(T_k^b - T_k^t\right)} \\
\overline{\left(T_2^b - T_2^t\right)\left(T_1^b - T_1^t\right)} & \cdots & \overline{\left(T_2^b - T_2^t\right)\left(T_k^b - T_k^t\right)} \\
\vdots & & \vdots \\
\overline{\left(T_k^b - T_k^t\right)\left(T_1^b - T_1^t\right)} & \cdots & \overline{\left(T_k^b - T_k^t\right)\left(T_k^b - T_k^t\right)}
\end{pmatrix}
\tag{3}
$$

In Equation (3) $T_k^b$ represents the temperature of the kth initial value, $T_k^t$ represents the temperature of the kth true value, and $\overline{\left(T_k^b - T_k^t\right)\left(T_k^b - T_k^t\right)}$ represents the covariance complex between the initial value temperature and the actual value temperature at the kth point.

b is the observed field covariance matrix.

$$
b = \begin{pmatrix}
\overline{\left(T_1^{obs} - T_1^t\right)\left(T_1^{obs} - T_1^t\right)} & \cdots & \overline{\left(T_1^{obs} - T_1^t\right)\left(T_k^{obs} - T_k^t\right)} \\
\overline{\left(T_2^{obs} - T_2^t\right)\left(T_1^{obs} - T_1^t\right)} & \cdots & \overline{\left(T_2^{obs} - T_2^t\right)\left(T_k^{obs} - T_k^t\right)} \\
\vdots & & \vdots \\
\overline{\left(T_k^{obs} - T_k^t\right)\left(T_1^{obs} - T_1^t\right)} & \cdots & \overline{\left(T_k^{obs} - T_k^t\right)\left(T_k^{obs} - T_k^t\right)}
\end{pmatrix}
\tag{4}
$$

In Equation (4), $T_k^{obs}$ represents the temperature of the kth observation, $T_k^t$ represents the temperature of the kth true value, and $\overline{\left(T_k^{obs} - T_k^t\right)\left(T_k^{obs} - T_k^t\right)}$ represents the covariance complex between the observed temperature and the true temperature at the kth point.

$c_i$ is the vector of the ith background covariance.

$$
c_i = \begin{pmatrix}
\left(T_1^b - T_1^t\right)\left(T_i^b - T_i^t\right) \\
\left(T_2^b - T_2^t\right)\left(T_i^b - T_i^t\right) \\
\cdots \\
\left(T_k^b - T_k^t\right)\left(T_i^b - T_i^t\right)
\end{pmatrix}
\tag{5}
$$

Equation (5) is the covariance matrix between the initial temperature and the true value of the column where the point to be fused is located.

Since the aim of data fusion in this paper is to obtain optimal results, this paper first carries out error assessment for both ERA5 and MERRA-2 in sub-regional sub-pressure layers, and uses the assessment results for data fusion. For example, when ERA5 is highly accurate, ERA5 is used as $T_i^{obs}$ to correct the $T_i^b$ of MERRA-2; when MERRA-2 is more accurate, ERA5 is $T_i^b$ and MERRA-2 is $T_i^{obs}$.

### 2.3.2. Evaluation Metrics

In this study, the Mean Absolute Error (MAE), Root Mean Square Error (RMSE), and Correlation Coefficient (R) are used to evaluate the accuracy of the fused data. The MAE is the average of the absolute values of the deviations between the fused results and the true values, which can be used to determine the degree of deviation of the fused results from the sounding data. The RMSE is the square root of the ratio of the square of the deviation between the fused result and the sounding data to the number of observations, which can measure the deviation between the fused result and the sounding data; the RMSE is more sensitive to outliers in the calculated data. The R can reflect the closeness between the fused results and the sounding data. The closer the RMSE and MAE are to 0, the higher the accuracy of the results, and the closer the R is to 1, the higher the correlation between them.

The three are calculated as follows:

$$\text{MAE} = \frac{1}{N} \sum_{i=1}^{n} |Y_i - X_i| \tag{6}$$

$$\text{RMSE} = \sqrt{\frac{1}{N} \sum_{i=1}^{n} (Y_i - X_i)^2} \tag{7}$$

$$R = \frac{\sum_{i=1}^{n} (Y_i - \overline{Y})(X_I - \overline{X})}{\sqrt{\sum_{i=1}^{n} (Y_i - \overline{Y})^2 \sum_{i=1}^{n} (X_i - \overline{X})^2}} \tag{8}$$

Equations (6)–(8) are the formulae used to obtain the MAE, RMSE, and R, respectively, where $Y_i$ represents the value of the fused result, $X_i$ represents the value of RAOB sounding data, and n represents the number of data samples. Before the accuracy evaluation, the spatial and temporal matching of the sounding data and the fused results are required.

## 3. Fusion of Temperature Profile Data

### 3.1. Data Fusion Process

In this paper, we take full advantage of the high observation accuracy of RAOB sounding data, combine the advantages of the horizontal resolution of ERA5 and the vertical distribution of MERRA-2, and adopt the optimal interpolation method to fuse the data of each pixel in order to obtain a fused product with high spatial resolution and accuracy. The specific operation is to extend the insufficient horizontal resolution of MERRA-2 to 0.25° resolution using the interpolation method. The data presented show that, although the vertical distribution of MERRA-2 is finer than that of ERA5, some of ERA5's pressure layers are not available for MERRA-2. The empirical equation between the temperature of the unknown pressure layer and the temperature of the known pressure layer is solved using both MERRA-2 and ERA5 so that the pressures of ERA5, RAOB, and MERRA-2 are all upgraded to 45 layers. In addition, for the true value, the accuracies of ERA5 and MERRA-2 are first verified using the RAOB with different regional and pressure layers, and the one with better accuracy is taken as the accurate and reasonable initial condition (initial field) under different conditions. Finally, ERA5 and MERRA-2, with 45 pressure layers and 0.25° horizontal resolution, are used for the optimal interpolation method calculation in order to obtain the final fused results, the flowchart for which is shown in Figure 2.

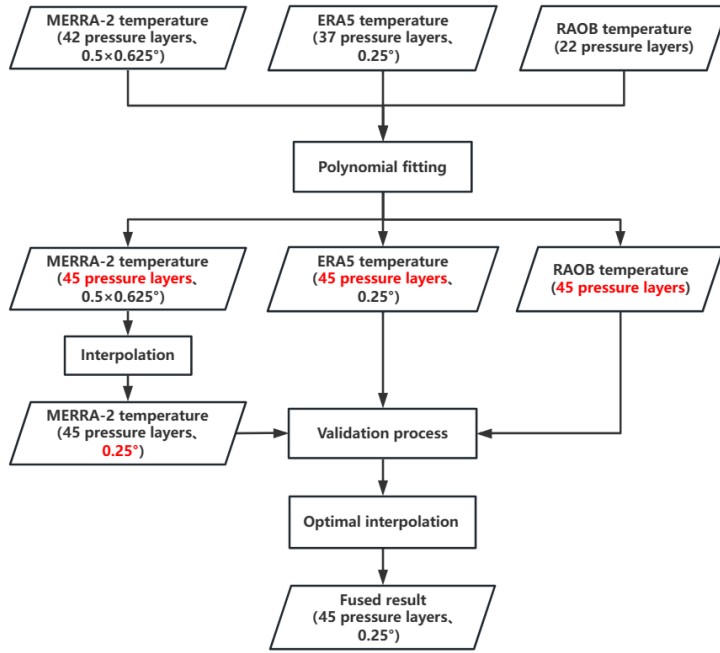

**Figure 2.** Temperature profile data fusion flowchart.

### 3.2. Regional Division

Firstly, by reading the data, it was found that most of the RAOB sounding data are detected twice a day, and the data times obtained are UTC 0700 and UTC 1900. In this study, the RAOB sounding data and ERA5 temperature profile data at UTC 0700 are downloaded, since the temporal resolution of the MERRA-2 temperature profile is 3 h. MERRA-2 data are not available at UTC 0700, so UTC 0600 is temporarily used instead. To ensure that the data in each region are optimally fused, the experiments calculated the accuracy of ERA5 and MERRA-2 temperature profiles in different regions for all seasons in 2019. Based on the experimental results, the globe was divided into regions as shown in Figure 3. The black dots in this figure show the distribution of some of the RAOB sounding balloon sites around the world, and it can be seen from the figure that the sounding sites are not continuous. However, they are distributed in various locations around the world.

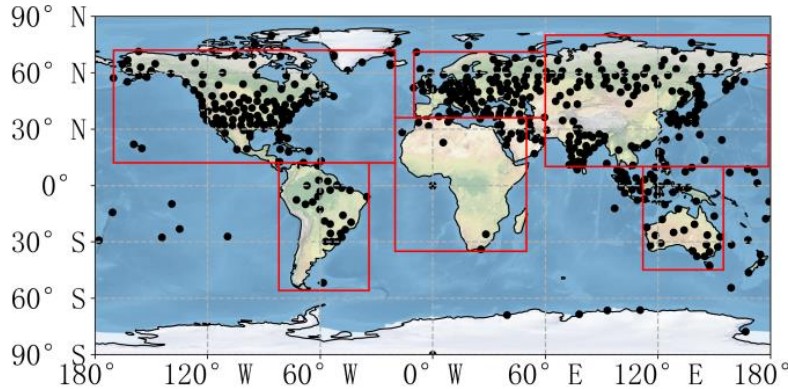

**Figure 3.** RAOB sounding data: global distribution of sites and area divisions.

### 3.3. Data Evaluation

In order to fuse ERA5 and MERRA-2 optimally, a preliminary evaluation of the data was first carried out. Based on the optimal interpolation method, the data with higher accuracy are used to correct the data with relatively poor accuracy, thus achieving the result of optimal data fusion. MERRA-2 and ERA5 were matched to data from RAOB, and

error calculations were conducted for each point, averaging the errors within each region according to the results of the regional division in Figure 2, and then analyzed. The results are shown in Tables 1 and 2. The analysis found that the accuracy of ERA5 is higher than that of MERRA-2 in most regions and seasons; the blue-bold font indicates the regions where the accuracy of MERRA-2 is better. To ensure that the optimal fusion is obtained for each pressure layer, the experiment further explored the errors of different pressure layers over different seasons, as shown in Figure 4, which shows the error profiles in winter 2019, where columns 1 and 3 represent the RMSE of ERA5, and columns 2 and 4 represent the RMSE of MERRA-2. The horizontal lines in each figure represent the data error, and the longer horizontal line represents the larger error. It is found that ERA5 and MERRA-2 have their advantages in different pressure layers. Therefore, this paper will use different correction methods for different pressure layers and regions. For the pressure layers where ERA5 demonstrates a higher accuracy, MERRA-2 will be used as the initial value, and ERA5 will be used as the observed value with which to correct MERRA-2. Moreover, for the pressure layers where MERRA-2 demonstrates a higher accuracy, ERA5 is used as the initial value and MERRA-2 as the observed value with which to correct ERA5.

**Table 1.** RMSE of MERRA-2 and ERA5 in different regions for all seasons (unit: K).

| | MERRA-2 | ERA5 | MERRA-2 | ERA5 | MERRA-2 | ERA5 | MERRA-2 | ERA5 |
|---|---|---|---|---|---|---|---|---|
| | (Spring) | | (Summer) | | (Autumn) | | (Winter) | |
| Asia | 8.72 | 7.00 | 12.68 | 3.57 | 7.87 | 8.23 | 7.71 | 8.31 |
| Europe | 10.89 | 9.18 | 16.65 | 2.19 | 12.03 | 9.32 | 9.80 | 8.47 |
| Oceania | 5.36 | 4.55 | 6.79 | 2.92 | 3.70 | 3.58 | 4.65 | 4.27 |
| Africa | 5.70 | 5.92 | 7.65 | 2.21 | 4.21 | 4.61 | 5.50 | 13.57 |
| North America | 9.46 | 8.19 | 15.30 | 2.45 | 8.35 | 7.49 | 8.70 | 9.61 |
| South America | 4.79 | 3.53 | 4.77 | 2.00 | 6.55 | 5.22 | 5.07 | 3.97 |

**Blue** for higher accuracy of MERRA-2.

**Table 2.** MAE of MERRA-2 and ERA5 in different regions for all seasons (unit: K).

| | MERRA-2 | ERA5 | MERRA-2 | ERA5 | MERRA-2 | ERA5 | MERRA-2 | ERA5 |
|---|---|---|---|---|---|---|---|---|
| | (Spring) | | (Summer) | | (Autumn) | | (Winter) | |
| Asia | 6.62 | 5.29 | 9.88 | 2.10 | 5.40 | 6.01 | 5.72 | 6.26 |
| Europe | 9.82 | 8.03 | 15.01 | 1.66 | 10.31 | 8.01 | 8.65 | 7.22 |
| Oceania | 3.70 | 3.19 | 4.59 | 1.91 | 2.82 | 2.65 | 3.37 | 3.11 |
| Africa | 4.89 | 5.06 | 6.84 | 1.83 | 3.50 | 3.86 | 4.66 | 4.79 |
| North America | 7.74 | 6.74 | 12.65 | 1.72 | 6.45 | 5.96 | 6.92 | 7.94 |
| South America | 3.88 | 2.79 | 3.62 | 1.72 | 5.98 | 4.54 | 4.24 | 3.27 |

**Blue** for higher accuracy of MERRA-2.

### 3.4. Data Fusion

The basic steps for the data fusion described in this paper include the following:

(1) Using existing data, a look-up table for converting known pressure layer temperatures to unknown pressure layer temperatures was calculated, and the datasets were converted to 45 pressure layers.

(2) The MERRA-2 temperature profile data were populated with global 0.25° data using interpolation.

(3) Based on the RAOB temperature data for all seasons in 2019, the accuracies of ERA5 and MERRA-2 were evaluated in different regional and pressure layers. Thereafter, the optimal fusion was applied to different pressure layers and regions.

(4) Algorithm results were evaluated for accuracy using RAOB sounding data.

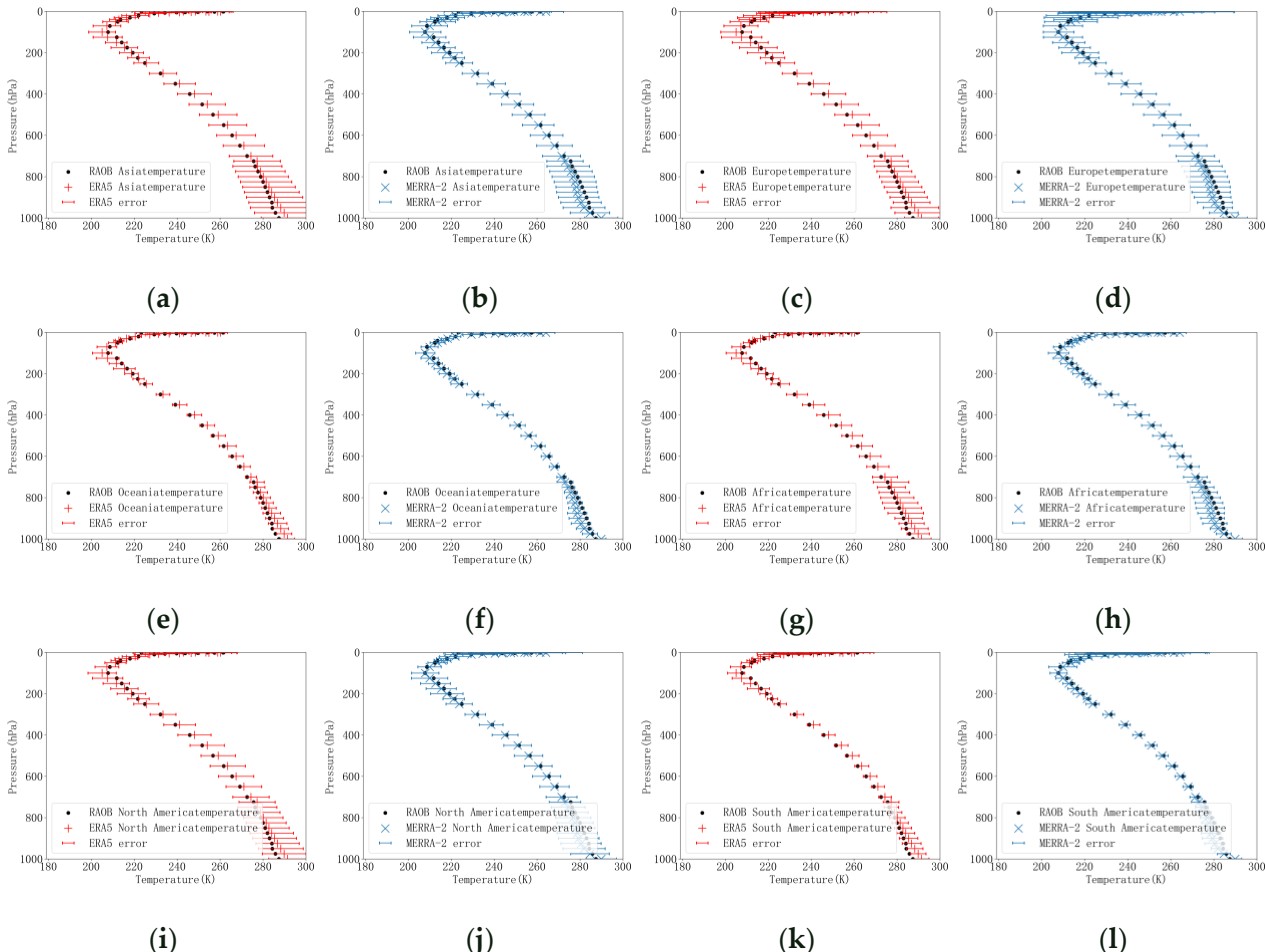

**Figure 4.** RMSE of MERRA-2 and ERA5 in winter 2019: (**a**) ERA5 (Asia), (**b**) MERRA-2 (Asia), (**c**) ERA5 (Europe), (**d**) MERRA-2 (Europe), (**e**) ERA5 (Oceania), (**f**) MERRA-2 (Oceania), (**g**) ERA5 (Africa), (**h**) MERRA-2 (Africa), (**i**) ERA5 (North America), (**j**) MERRA-2 (North America), (**k**) ERA5 (South America), and (**l**) MERRA-2 (South America).

By investigating the vertical distribution of the pressure recorded using ERA5 and MERRA-2, it was found that the merged set of both should have 45 pressure levels: 1000, 975, 950, 925, 900, 875, 850, 825, 800, 775, 750, 725, 700, 650, 600, 550, 500, 450, 400, 350, 300, 250 225, 200, 175, 150, 125, 100, 70, 50, 40, 30, 20, 10, 7, 5, 4, 3, 2, 1, 0.7, 0.5, 0.4, 0.3, and 0.1 (hPa); therefore, the corresponding look-up tables were created by polynomially fitting the pressure layer at the known temperature to the pressure layer at the unknown temperature using two datasets—those of ERA5 and MERRA-2—so that, for the corresponding look-up tables, all sets are represented by equations for y $= ax^2 + bx + c$, applied to the datasets of RAOB, ERA5, and MERRA-2, in order to find the data corresponding to the pressure layer of the unknown temperature, and, finally, the vertical distribution of all datasets was extended to 45 layers for subsequent calculations.

In order to validate the accuracy of this method, the RAOB data stored at UTC 0000 on 1 January 2020 were read. The real time of the data was calculated to be UTC 0700, so the temperature profile data of the datasets at the Beijing site (116°20′E, 39°56′N) at UTC 0700 on 1 January 2020 were read and plotted. In Figure 5a, the temperature profiles of MERRA-2, ERA5, and RAOB are shown.

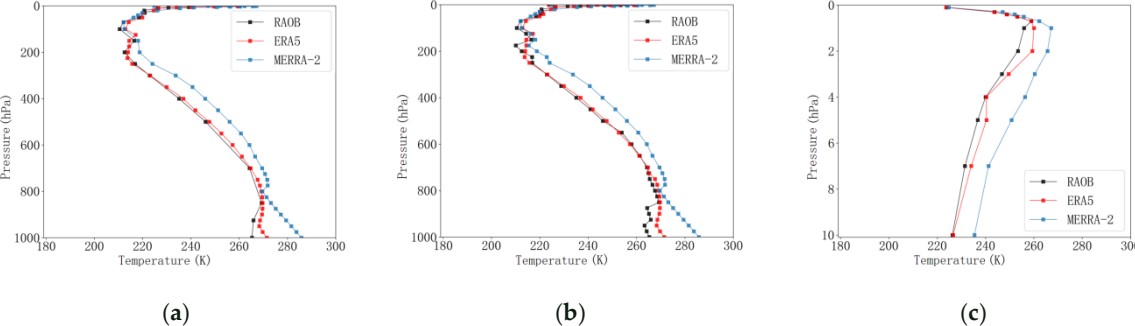

**Figure 5.** Temperature profiles of MERRA-2, ERA5, and RAOB in the Beijing area: (**a**) initial temperature profile, (**b**) temperature profile after preprocessing, and (**c**) mesosphere temperature profile after preprocessing.

After processing using the look-up table method, the temperature profiles of the datasets in the Beijing area at UTC 0700 on 1 January 2020 were drawn again for comparison, as shown in Figure 5b. It was found that the temperature profile results generated using this method were in normal order, and, therefore, all temperature profile data were converted to 45 pressure layers using the empirical formulae obtained. The temperature profiles of the datasets in the middle layer in Beijing are shown in Figure 5c, and their shapes are also roughly similar.

The subsequent use of the RAOB sounding data included the processing into 45 pressure levels, to verify the ERA5 and MERRA-2 temperature profile data at the same time and with the same number of layers. The results are shown in Figure 6a, where the blue line represents the error of MERRA-2, and the red line represents the error of ERA5. By calculation, the mean RMSE of the ERA5 temperature profile is approximately 5.7 K, and the mean MAE value is approximately 4.3 K. The mean RMSE value of MERRA-2 temperature profile is approximately 11.7 K, and the mean MAE value is approximately 8.9 K. The mean values of both the RMSE and MAE of ERA5 are approximately 5 K smaller than those of MERRA-2. Figure 6b shows the error of the temperature profile in the middle layer. The mean RMSE value of the ERA5 temperature profile is approximately 7.8 K, and the mean MAE value is approximately 5.9 K. The mean RMSE value of the MERRA-2 temperature profile is approximately 7.9 K, and the mean MAE value is approximately 5.9 K. The error in the mesosphere is large compared to that in the troposphere, probably because the true value in the mesosphere is calculated by the look-up table, which exhibits some uncertainty.

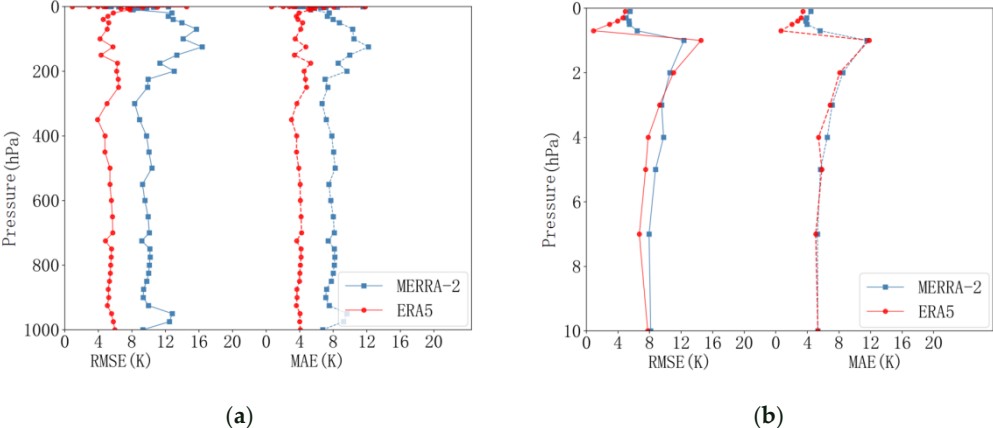

**Figure 6.** Error of MERRA-2 and ERA5 temperature profiles after preprocessing. (**a**) Error. (**b**) Mesospheric error.

Since the spatial resolution of MERRA-2 is $0.5° \times 0.625°$, it needs to be interpolated to $0.25°$. After interpolation, the horizontal resolution of both MERRA-2 and ERA5 is

0.25°. Finally, the two datasets, with a horizontal resolution of 0.25° and a pressure layer of 45 layers, were substituted into the optimal interpolation method formula described in Section 2.3.1 in order to calculate the fused product results.

## 4. Results and Analysis

### 4.1. Results of Data Fusion

After fusing the two types of data using the above method, RAOB was compared with the temperatures corresponding to the matching points of the fused results, and the RMSE of each point was calculated, in order to judge the algorithm's result accuracy effectively.

The two columns on the left side of Figure 7 show the values of the partial pressure layers of the RAOB temperature data and the values of the corresponding pressure layers of fused temperature data. Both sets of data were first processed to remove the invalid values from each pressure layer of the RAOB sounding data, then to determine the temperature at the station corresponding to the values of the fused data, and, finally, to draw the spatial distribution map of both after matching. As shown in the picture, the results of the latest algorithm are similar to the spatial distribution of RAOB data in different pressure layers. However, as some sites in North America and Asia have larger errors, the preliminary judgment is that the fused data still contain some errors. The RMSEs of the fused results for each point were calculated and presented in the right column of Figure 7, where the maximum value is set to 10 K. The closer the color is to red, the larger the error is, and the closer it is to blue, the smaller the error is. It can be seen that the errors of individual points are still large, and the subsequent optimization of the algorithm can be considered to further progress.

### 4.2. Evaluating the Accuracy of Fused Data

In order to further quantitatively judge the accuracy of the algorithm results, error calculations and correlation tests were performed on the fused results using the RAOB sounding data. The results were obtained by averaging the errors using the global sites as shown in Figure 8.

As shown in Figure 8a, the accuracy of the fused data based on ERA5 and MERRA-2 is calculated as the RMSE and MAE, from left to right. The RMSE of the temperature profile of the pressure layers below 100 hPa is within 8 K, and the MAE is within 6 K. The errors and R before and after fusion of the ERA5 and MERRA-2 data are summarized in Table 3, which shows that the RMSE and MAE of MERRA-2 were reduced by 6.3 K and 5.0 K, respectively, and the RMSE and MAE of ERA5 were reduced by 0.3 K and 0.4 K, respectively. The R for both datasets also improved to some extent.

In addition, the algorithm-processed temperature profiles have a horizontal resolution of 0.25° and a vertical distribution of 45 layers, compared to 0.25° and 37 layers for ERA5, and 0.5° × 0.625° and 42 layers for MERRA-2. The resolution of the fused data is much finer. Figure 8b shows the mesosphere layer temperature profile error plot after data fusion. Additionally, after data fusion, the mean value of the RMSE of the mesosphere layer temperature profile is 6.7 K, which is reduced by approximately 1.1 K compared with that of ERA5. The mean MAE value of the temperature profile is 5.0 K, which is reduced by approximately 0.9 K compared with that of ERA5. The mesosphere layer error is also reduced. Figure 8c shows the R of the data fusion with the RAOB temperature profile, and it can be found that the correlation between the two datasets is already very close to 1. However, there are still some differences. By comparing the R before data fusion, it can be seen that some overestimated points in the scatter plot of the fusion results display consistency with ERA5, and it can be assumed that ERA5 has some influence on the error in data fusion. The points with large errors can be considered separately for subsequent studies in order to explore the reasons for the large errors and correct them.



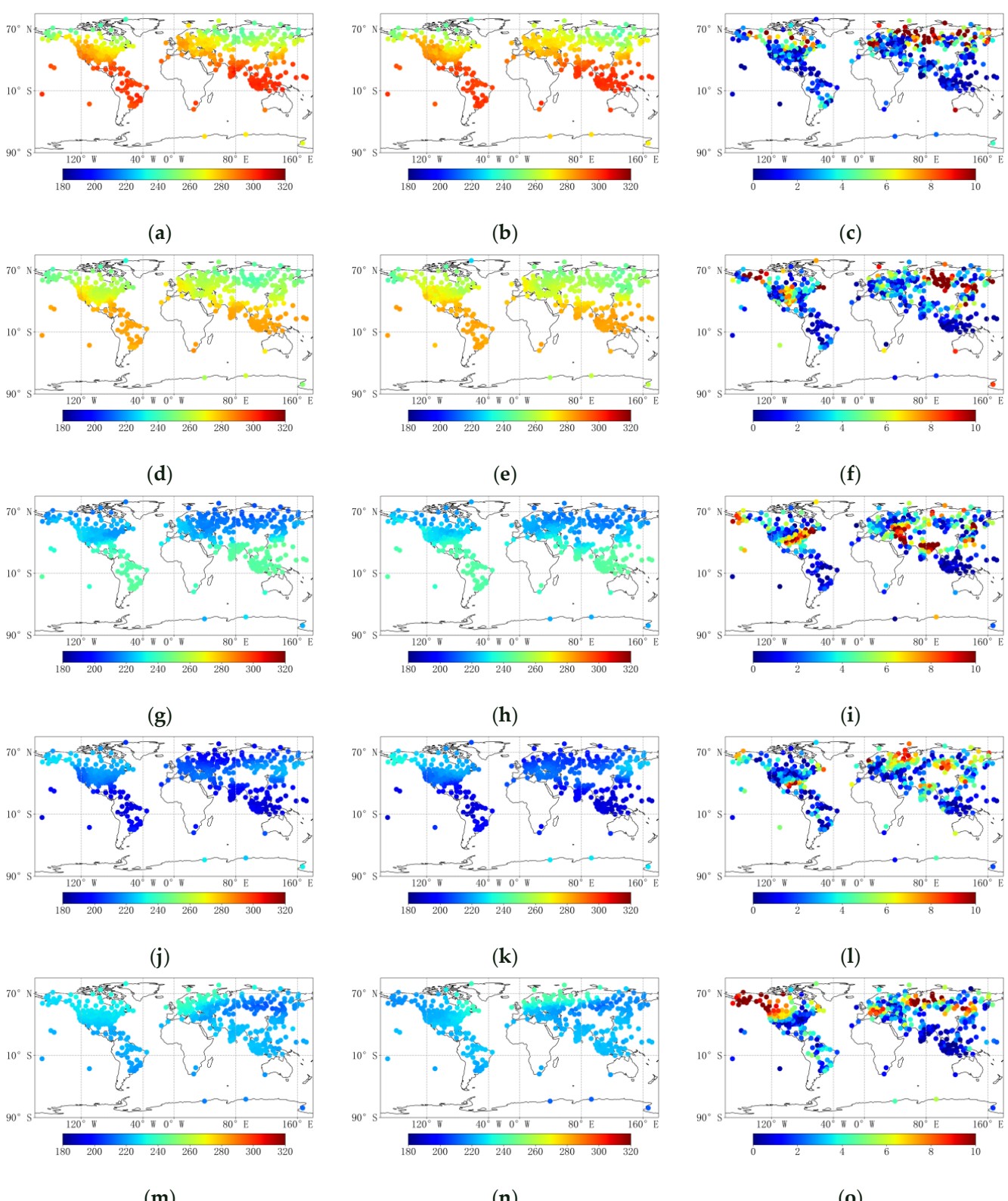

**Figure 7.** Values and RMSE(K) for the same location of RAOB temperature and fused temperature at different pressure layers. (**a**) RAOB (1000 hPa). (**b**) Fused result (1000 hPa). (**c**) RMSE(K) (1000 hPa). (**d**) RAOB (700 hPa). (**e**) Fused result (700 hPa). (**f**) RMSE(K) (700 hPa). (**g**) RAOB (300 hPa). (**h**) Fused result (300 hPa). (**i**) RMSE(K) (300 hPa). (**j**) RAOB (100 hPa). (**k**) Fused result (100 hPa). (**l**) RMSE(K) (100 hPa). (**m**) RAOB (0.1 hPa). (**n**) Fused result (0.1 hPa). (**o**) RMSE(K) (0.1 hPa).

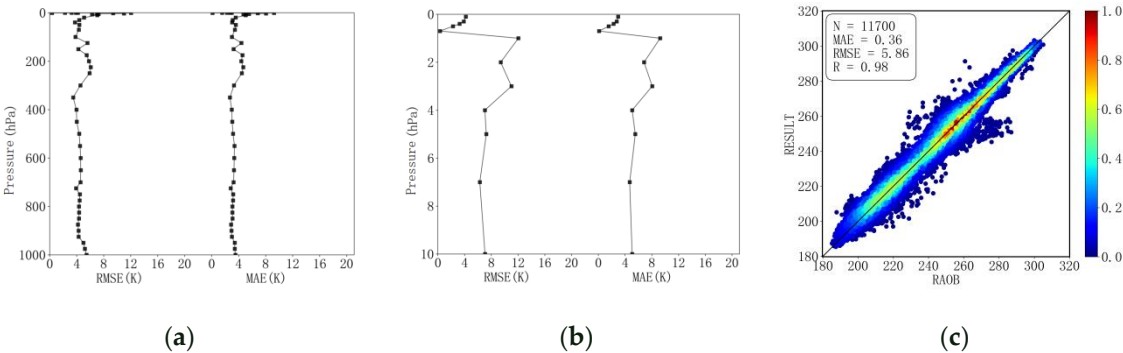

(**a**)　　　　　　　　　　(**b**)　　　　　　　　　　(**c**)

**Figure 8.** Fused temperature profile error and its correlation with RAOB. (**a**) Error. (**b**) Mesospheric error. (**c**) Scatter Chart.

**Table 3.** Error and R before and after data fusion.

|  | RMSE(K) | MAE(K) | R |
| --- | --- | --- | --- |
| 45 pressure layers MERRA-2 | 11.7 | 8.9 | 0.92 |
| 45 pressure layers ERA5 | 5.7 | 4.3 | 0.95 |
| Fused data | 5.4 | 3.9 | 0.98 |

### 4.3. Validation of Fused Data

In order to prove that the algorithm is generalized for all seasons, data from all seasons of 2019 were substituted into the algorithm for validation, and the accuracy of the fused data was higher than the accuracy of the reanalysis data before processing. Figure 9 shows the correlation between the fused temperature and RAOB sounding data for all seasons of 2019, from top to bottom: the scatter plot of ERA5, that of MERRA-2, and that of the fused data. It can be observed that the fused results are more correlated with the RAOB sounding data.

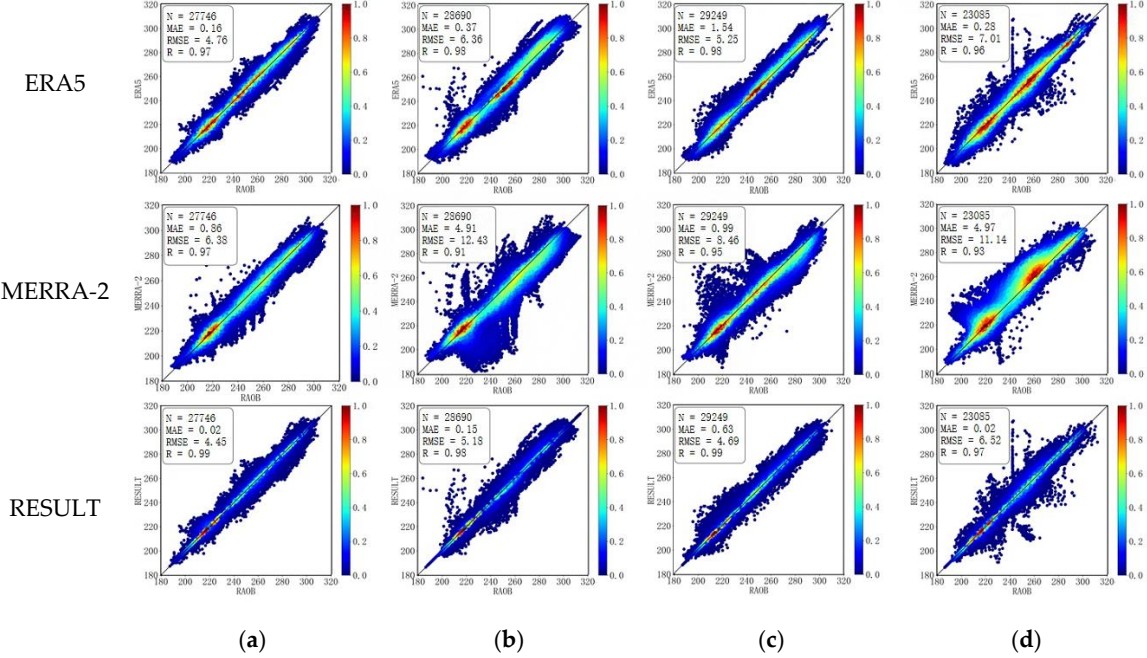

(**a**)　　　　　　　　　(**b**)　　　　　　　　　(**c**)　　　　　　　　　(**d**)

**Figure 9.** Fused data and RAOB correlation in the four seasons of 2019. (**a**) spring. (**b**) summer. (**c**) autumn. (**d**) winter.

The temperature profile fused algorithm in this study can be applied globally. At each pixel point, RAOB, ERA5, and MERRA-2 atmospheric temperature profile data are input, and the algorithm is used to fuse the data, resulting in a global atmospheric temperature profile product with a horizontal resolution of 0.25° and a vertical distribution of 45 pressure levels. The validation results show that the fused algorithm's product results are highly accurate and can provide important data for the support of subsequent meteorological studies.

## 5. Conclusions

In this paper, we proposed a fusion method to yield high spatiotemporal atmospheric temperature profiles from RAOB, ERA5 and MERRA-2 data based on the optimal interpolation method. The method takes advantage of the high observation accuracy of RAOB sounding data, combines the advantages of the horizontal resolution of ERA5 and the vertical distribution of MERRA-2, and adopts the optimal interpolation method in order to fuse the data with a horizontal resolution of 0.25°, a vertical distribution of 45 pressure levels, and high accuracy. The following conclusions were drawn:

A polynomial fitting method was used to develop an empirical formula for converting known pressure layer temperatures to unknown pressure layer temperatures, allowing the data to be filled with 45 pressure layers.

The accuracy is reduced by 6.0 K for RMSE and 5.0 K for MAE relative to the MERRA-2 temperature profile, and by 0.3 K for RMSE and 0.4 K for MAE relative to the ERA5 temperature profile.

By comparing the values of the fused data with the RAOB sounding data and the RMSE(K) at different stations, it was found that the fused data was already very close to the RAOB sounding data. Additionally, the fused data was the finest, in terms of both horizontal resolution and vertical distribution, of any directly downloadable product available. Therefore, this method has the potential to be applied to meteorological studies in order to provide finer temperature profile products.

Since the time difference between the MERRA-2 initial temperature profile data and RAOB is one hour, this will affect the accuracy of the final fused product. Therefore, we will consider interpolating the MERRA-2 data in the future, and then taking the ERA5 and MERRA-2 data at the same time into the algorithm, so that hour-by-hour fused results can be calculated. In addition, using the current fused results, the temperature profile can only be verified against the daily RAOB data at the same moment, and in the future, the RAO丨combined with ground-based microwave radiometer observations can be evaluated in order to validate the accuracy of the fused data.

**Author Contributions:** Y.Q.: conceptualization, data curation, methodology, formal analysis, and writing—original draft. D.J.: conceptualization, methodology, writing—original draft, writing—review and editing, and funding acquisition. H.S.: conceptualization, supervision, investigation, writing—review and editing, and funding acquisition. J.X.: supervision and project administration. R.X.: validation and visualization. C.S.: formal analysis, visualization, and writing—review and editing. All authors have read and agreed to the published version of the manuscript.

**Funding:** This research was funded by: Second Tibetan Plateau Scientific Expedition and Research Program (STEP) 2019QZKK0206. National Science Foundation of China, under Project no. 42175152. Youth Innovation Promotion Association, under Project no. 2021122.

**Data Availability Statement:** The European Centre for Medium-Range Weather Forecasting (ECMWF) ERA5 data, presented in this study, are openly available at (https://cds.climate.copernicus.eu/#!/search?text=era5, last accessed on 10 May 2023) at (DOI: 10.24381/cds.f17050d7). The Modern-Era Retrospective Analysis for Research and Applications, Version 2 (MERRA-2) data, provided in this study, are openly available at (https://disc.gsfc.nasa.gov/, last accessed on 10 May 2023). The radio sounding observational data, presented in this study, are openly available at (https://ruc.noaa.gov/raobs/, last accessed on 10 May 2023).

**Acknowledgments:** The authors are very grateful to the NOAA/ESRL Radiosonde Database for making the data available. We thank the European Centre for ERA5 (the fifth-generation ECMWF Re-

analysis) and NASA's MERRA-2 (Modern-Era Retrospective Analysis for Research and Applications, version 2) for making the datasets available. This work was supported by Second Tibetan Plateau Scientific Expedition and Research Program (STEP) 2019QZKK0206.

**Conflicts of Interest:** The authors declare no conflict of interest.

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
