# Peer review of "The Fusion of ERA5 and MERRA-2 Atmospheric Temperature Profiles with Enhanced Spatial Resolution and Accuracy"

_remotesensing, doi:10.3390/rs15143592_

Round 1

Reviewer 1 Report

The paper addresses the interesting topic of the atmospheric temperature profile data fusion. It considers two huge datasets (ERA5 and MERRA-2) as basis to produce a consistent global dataset. Raobs data are employed as ground truth to validate the quality of the merged data.

The flowchart depicted in Figure 1 (that comes after Figure 2 in the main body of the text!!!) clearly explain the fusion  process. Unfortunately, the manuscript does not describe clearly some essential passages. 

In Sec 2.3.2 (Region division) the spatial distribution  map of error "showed some consistency" , which is a qualitative judgement that has no scientific sound. Sec 2.3.3 (Data fusion methods)  should be rewritten, as it is not clear what the background is or represent in comparison to the observed (?) and analyzed(?) values. Are observation those that comes from Raobs? The temperature to be fused is the final value, that is the results of the process? I was convinced that the result would be Tia

In addition, the notation is confusing as matrices are usually indicated by Bold Capital letters.

Tables 1 and 2 of sec 3.1 (Data evaluation) reports global values without specific comments. I do not understand the physical  meaning of a mean temperature profile value referred to such an area as Asia, for instance...It seems to me that some assumptions are missing. That is why , I find difficult to understand the quality of the numerical results.

I would appreciate that the authors refine the rationale of their paper and resubmit it to help the reviewer in expressing his/her recommendation.

The manuscript needs a thorough revision as far the English usage is concerned.

Section1 ( Introduction) contains sentences whose meaning is weird.

Just few examples...

Pag 2:

"So it is also important to observe or predict the temperature of the mesosphere atmosphere for weather studies such as the troposphere or surface (Wang, 2021)."

"Fengyun satellites, JPSS(Joint Polar Satellite System), and other satellites." (is this a sentence?)

"Airborne measurement is expensive and lacks time continuity, so this measurement method is generally not used; for ground-based measurement, including microwave radiometer, sounding balloon measurement, and other methods, ground based microwave radiometer has good time continuity for temperature profile observation, but it is greatly affected by weather, especially under cloudy conditions, the uncertainty of the cloud absorption coefficient leads to an increase in error or even failure." (In this sentence there is a mix of subjects, facts and comments. Please rewrite)

Pag 3

"It used the one-Dimensional Variational method based on the optimal interpolation method to finally obtain the fused product, which is relatively the initial separate products were improved to some extent (Gamage et al.,2020)." (Who or what is the subject of this sentence? Gamage et al...if yes it should they. Anyhow the meaning of the sentence is obscure)

Then, Figure 2 is cited in the main body of the text prior to Figure 1....

Reviewer 2 Report

The paper reports on the fusion of temperature profiles from MERRA-2 and ERA 5. RAOB data are used to validate the obtained results. In my opinion the paper must be improved before the publication both on the point of view of the English style (sometimes it is really difficult to understand the meaning of some phrases) but particularly for the presentation of the results.

Below some technical/specific comments. It is difficult to refer to specific  points in the paper since there is no line numeration, I've done my best to identify the single points.

Abstract

pag 1: The validation using data in 2019 show --> shows

pag 1: accuracythan --> typo: accuracy then

Introduction

pag 2: The phenomenon of temperature inversion ... phrase too long, please reformulate

pag 2: military importance: why military?

pag 2: mesosphere atmosphere --> remove atmosphere

pag 2: on board which --> on board, which

pag 2: MWHTS add acronym

pag 2: Airborne measurement --> new paragraph

pag 2: data accuracy validation, for example --> phrase too long add a point

in several points in the paper e.g. "Ma, Y., et al. used" add date of the paper

pag 2: significant platforms ? this phrase is not clear for me what does it means "significant platforms"?

pag 2: ship... --> the authors do not introduce ship measurements in the introduction

pag 2: Data assimilation --> new paragraph

pag 3: correct the bias of the data --> is data assimilation used to correct the biases? re-phrase please

pag 3: 3D/4D variational, and Kalman filters --> add references for these techniques

pag 3 are similar in that they both make the difference between the observed values and the points to be assimilated and interpolate them to the values of the points to be assimilated, and finally obtain the results as analytical values. The difference is that the optimal interpolation

method calculates the weight function by minimizing the analytical variance. --> Phrase too long and unclear reformulate

pag 3: method based on the optimal interpolation method to finally obtain the fused product, which is relatively the initial separate products were improved to some extent --> not clear reformulate

pag 3: Usually, the reanalysis data cover -->  the authors introduce satellite and other measurement dataset, while reanalysis are not introduced ... please add a phrase on this

pag 3: which can reach 1 , 0.25 --> typo?

pag 4: to optimally fuse the two data--> two data, MERRA and ERA?

pag 4: Radio soundings (radiosonde --> The authors did not quantify here the accuracy of RAOB on temperature profiles ... please add informations

pag 4: As shown in Figure 2 --> figure 2 is before Figure 1 change

pag 4: Reanalysis is the process of reprocessing ..--> This is an introduction, move in into the right section and then start here with ERA5.

pag 6: In this paper, we take -->  This a description of the method the authors use, it is a new section in materials and methods

pag 6: in Figure 1 --> figure 2 now

pag 6:  UTC 0700 and UTC 0600 --> Is there any consequence from this temporal mismatch? please discuss

pag 7: In figure 3 , which value of R are you using in this work?

pag 7: verification results in 3.1, the --> section 3.1

pag 9: The horizontal lines in each figure represent --> put the figure description in the legend and here leave only the considerations

pag 9: larger error. --> lager errors ?? what does it means

pag 10: figure 4 legend Error bar plots --> RMSE

pag 10: three data --> datasets

pag 10: so that The --> typo the

pag 10: to find the

data corresponding to the pressure layer of the unknown temperature, and finally, the vertical distribution of all three data is extended to 45 layers for subsequent calculations --> this phrase is too long and unclear

pag 10 To validate the accuracy of this method, read the RAOB data stored at UTC 0000 on

January 1, 2020. This phrase has something wrong re formulate

pag 10: three data --> datasets

pag 11: It can be seen that most points of ERA5 temperature

profile are closer to the RAOB sounding data than MERRA-2, but there are still some points where MERRA-2 is closer to RAOB. --> From these plots MERRA seem completely out! more than 10K differences! please add a comment on this... from these plots I do not see the motivation of using MERRA

pag 11: This is a comment: Figure 5 and 6 seems more results than methods

pag 13: As shown in Figure 8(a),-->  are these average results over the whole globe?

pag 13: The RMSE of the temperature profile of the pressure layer below 3 hPa is within 8 K, --> not clear, where are these values shown? in Figure 6?

pag 13: The mean value of RMSE of ERA5 temperature profile before processing is 5.7 K, which is reduced by about 0.3 K. The mean

value of MAE of ERA5 temperature profile before processing is 4.3 K ---> it is better to add a table with all these values

pag 13: the algorithm-processed temperature profile is more accurate

in terms of horizontal resolution and --> how do you evaluate this? add explanations

pag 13: By comparing the R before data fusion, it can be seen that ERA5 has a certain influence on the error in data

fusion. --> where can this be seen? add details

Pag 14: Figure.9 --> add in the legend what there is in each row

pag 14: It is not shown at which altitude levels MERRA-2 contributes, I think the authors should add a table on this because in the paper it is not fully clear the contribution of MERRA data and since the authors state they are relevant for altitude resolution, this should be evident in the paper.

pag 14: add a table summarizing the results

Conclusions

I think that in the conclusion section the authors should give some information on how these results impact on the general knowledge in the field (e.g. results vs requirements) and which are the possible applications of the obtained results.

See my comments in section above

Round 2

Reviewer 1 Report

The authors improved the quality of their manuscript as far as the structure of the paper and the usage of English are concerned.

My recommendation is to publish as is.

Reviewer 2 Report

I really appreciate the author's effort in addressing all my comments